# Correlation of ENT Symptoms with Age, Sex, and Anti-SARS-CoV-2 Antibody Titer in Plasma

**DOI:** 10.3390/jcm12020610

**Published:** 2023-01-12

**Authors:** Aleksandra Kwaśniewska, Krzysztof Kwaśniewski, Andrzej Skorek, Dmitry Tretiakow, Anna Jaźwińska-Curyłło, Paweł Burduk

**Affiliations:** 1Department of Otolaryngology, Laryngological Oncology and Maxillofacial Surgery, University Hospital No. 2, 85-168 Bydgoszcz, Poland; 2Department of Vascular Surgery and Angiology, University Hospital No. 1, 85-094 Bydgoszcz, Poland; 3Department of Otolaryngology, Medical University of Gdańsk, 80-210 Gdańsk, Poland; 4Regional Center of Blood Donation and Treatment, 80-210 Gdańsk, Poland; 5Department of Otolaryngology, Phoniatrics and Audiology, Collegium Medicum, Nicolaus Copernicus University, 85-168 Bydgoszcz, Poland

**Keywords:** COVID-19, convalescent plasma, SARS-CoV-2, ENT symptoms

## Abstract

Our objective is to evaluate the correlation between ENT symptom occurrence and antibody titer in convalescent plasma, as well as the influence of age and gender on ENT manifestations of COVID-19. We measured the levels of antibodies in 346 blood donors, who had PCR-confirmed previous infection and met the study inclusion criteria. We recorded otolaryngological symptoms during infection: dry cough, dyspnea, sore throat, smell/taste disturbances, vertigo, dizziness, nausea and vomiting, sudden unilateral loss of hearing, progressive loss of hearing, and tinnitus. In addition, we statistically analyzed the correlation between patients’ antibody levels, symptoms, age, and gender using a chi-square test or Fisher exact test. A *p*-value less than 0.05 determined statistical significance. The mean age of the convalescents was 39.8 ± 9.56 SD and the median of the measured anti-SARS-CoV2 plasma antibodies was 1:368.5. The most common ENT symptoms were smell/taste disturbances (62.43%), dry cough (40.46%), sore throat (24.86%), and dyspnea (23.7%). Smell and taste disturbances were more frequent in younger patients and the marked antibody titer was lower, which was contrary to a higher antibody titer associated with dry cough, dyspnea, and dizziness. Occurrence of sore throat was not correlated with age, sex, or antibody level. There were no significant differences in otological symptoms in female patients. Gender does not affect the occurrence of ENT symptoms. The symptomatic course of SARS-CoV-2 infection is not always associated with higher levels of antibodies in the blood. The age of the infected patients, unlike gender, affects the occurrence of some ENT symptoms.

## 1. Introduction

Coronavirus disease 2019 (COVID-19) is an ongoing problem caused by severe acute respiratory syndrome coronavirus 2 (SARS-CoV-2). Initial infection appeared in Wuhan (China) in December 2019 and was declared a pandemic by the World Health Organization (WHO) in March 2020. According to the WHO, COVID-19 has infected over 600 million people and caused over 6.5 million deaths worldwide [1].

Symptoms of the disease are varied; besides general manifestations such as fever or chills, ear, nose, and throat (ENT) symptoms are also prevalent [2]. According to various authors, ENT symptoms associated with COVID-19 could include sore throat, headache, cough, dyspnea, pharyngeal erythema, nasal congestion and obstruction, rhinorrhea, nasal itching, upper respiratory tract infection, tonsil enlargement, sneezing, dysphagia, voice impairment, olfactory and taste dysfunction, dizziness, vertigo, tinnitus, and hearing impairment [2,3,4,5].

However, their frequencies differ in patients, as do anti-SARS-CoV-2 antibody levels in their plasma [6]. Treatment with convalescent plasma rich in anti-SARS-CoV-2 antibodies has proven effective in reducing the severe course and mortality of COVID-19 [7]. The correlation of demographic factors and the level of antibodies in relation to their effects on disease symptomatology has been evaluated previously [8,9,10,11]. However, only correlation with the severity of the disease was tested, without stratification by each symptom. Furthermore, it was investigated only for the general symptoms of COVID-19 infection, and not with a specific focus on otorhinolaryngological symptoms.

As has been shown by other authors, the severity of the disease affects anti-SARS-CoV2 antibody levels [12,13,14]. We aimed to analyze if particular otolaryngological symptoms could have a similar predictive value. Investigating the relationship between ENT symptoms and the level of antibodies may facilitate the prediction of the severity, duration and complications of COVID-19 based on the presence or absence of respective symptoms. This research may help otolaryngologists identify patients with COVID-19 infection, and, after further research, predict the course of infection. This could have both scientific and clinical significance.

This study aimed to show the frequency of ENT symptoms among COVID-19 patients, their occurrence depending on sex and age, and the correlation with IgG anti-SARS-CoV-2 antibody titers in convalescent plasma.

## 2. Materials and Methods

COVID-19 convalescents (n = 346) who donated blood at the Regional Center of Blood Donation and Treatment in Gdańsk (Poland) were enrolled in the study. The objective of blood donation was to acquire plasma rich in anti-SARS-CoV-2 antibodies to be used for the treatment of severe COVID-19 cases. SARS-CoV-2 infection was confirmed by polymerase chain reaction (PCR) testing of nasopharyngeal swabs. The convalescents’ blood was donated from 10 to 120 days after a fourteen-day isolation period. None of the donors were vaccinated or hospitalized due to COVID-19. Patients were asked about otolaryngological symptoms of the disease: dry cough, dyspnea, sore throat, smell/taste disturbances, vertigo, dizziness, nausea and vomiting, sudden unilateral loss of hearing, progressive loss of hearing, and tinnitus. The inclusion criteria were confirmed SARS-CoV-2 infection, 18–64 years of age, and normal complete blood count, blood pressure, pulse, and body temperature. The exclusion criteria were autoimmune diseases, anti-HLA antibodies in the blood, active infection (including Treponema pallidum) or oncological illness, history of HIV, Hepatitis B or Hepatitis C infection, and being under the influence of psychoactive substances. The testing methodology of a study by Skorek et al. was replicated [11]. Blood tests were performed using the SARS-CoV-2 S-RBD IgG test of a MAGLUMI 800 device (Snibe Co., Shenzhen, China). Serological tests were performed using the in vitro chemiluminescent kit (Cat. No. SARS-CoV-2 S-RBD IgG122, Mindray, China) for the quantification of S-RBD IgG neutralizing antibodies (nAbs) against SARS-CoV-2. After collecting blood from the examined person, it was placed in test tubes with a separating gel or clot activator. After centrifugation (>10,000 RCF for 10 min), a sample (10 µL volume) containing no fibrin or other solids was collected. Subsequently, the sample, along with the buffer and magnetic particles coated with the recombinant S-RBD antigen, were mixed and incubated, resulting in the formation of immune complexes. After magnetic field precipitation, the supernatant was removed and washed. After the addition of ABEI-labeled anti-human IgG antibodies, the sample was subjected to another incubation and precipitation followed by washing to remove unbound proteins from the sample. Finally, the chemiluminescence reaction was initiated and the light signals were measured with a photomultiplier for 3 s in relative light units (RLUs), which are proportional to the SARS-CoV-2 S-RBD IgG concentration. In the case of a test performed 15 days after the onset of symptoms, the sensitivity of the test (according to the manufacturer) is 100.0% and its specificity is 99.6% (CE REF 30219017 M) [15]. Participation in the study was voluntary and written consent was obtained. The data collected were statistically analyzed using the chi-square test or Fisher exact test when the chi-square test assumption was not fulfilled (theoretical values in each cell of the contingency table equal to at least 5) to derive *p*-values. A post hoc Bonferroni correction was applied. A *p*-value less than 0.05 (typically ≤ 0.05) determined statistical significance. Cohen’s h effect size was calculated and classified as small (h = 0.20), medium (h = 0.50) or large (h = 0.80). These calculations were prepared using the Statistica 13.3 StatSoft PL software (Collegium Medicum, Nicolaus Copernicus University, Bydgoszcz, Poland) and the R statistical package version 4.0.2. (Biostat, Warsaw, Poland).

The study was approved by the Regional Independent Bioethics Committee, Gdansk Medical University, Poland (NKBBN 199/2021).

## 3. Results

The study included 302 males and 44 females, whose mean age was 39.8 ± 9.56 SD (age range 18–64) (Table 1). The median of the measured plasma IgG anti-SASR-CoV2 antibody titers was 1:368.5. Median rather than mean antibody titer levels were calculated to minimize outlier values and for a more accurate predictive value. The FDA recommends 1:160–1:640 as the standard antibody level, which coincides with our work (the mean in the standard range is 1:400). Patients were divided into groups depending on sex and plasma antibody level.

The most common ENT manifestations of COVID-19 infection were smell/taste disturbances (62.43% of patients), dry cough (40.46%), sore throat (24.86%), and dyspnea (23.7%). Others included in the study were vertigo (11.85%), dizziness (8.09%), tinnitus (6.07%), nausea and vomiting (3.76%), sudden unilateral loss of hearing (1.73%), and progressive loss of hearing (0.58%).

We noted a statistically significant correlation between otolaryngological symptoms manifested during COVID-19 infection and measured antibody levels. The occurrence of dry cough, dyspnea, and dizziness was associated with higher antibody titers (Figure 1; Table 2). Additionally, in male patients there was a positive correlation between dry cough, dyspnea, dizziness, vertigo, and higher antibody titers (*p* < 0.05). Interestingly, smell/taste disturbances were correlated with lower antibody titers (Figure 2 and Figure 3; Table 2).

Our study showed no statistically significant differences between sex and occurrence of any ENT symptom of COVID-19 (Table 2). An exception was seen in patients with lower antibody titers, whereby nausea and vomiting were more common in women than in men.

Patients were divided into younger and older groups based on mean age (under and above the age of 38.9). Smell/taste disturbances were more common in younger convalescents (statistically significant difference, *p*  <  0.05) (Table 2). Likewise, this difference was maintained both within the male and female patient distributions (*p* < 0.05). There were no statistically significant correlations between sore throat and age, sex, or antibody level. In addition, there was no correlation between antibody titers and ENT symptoms in women (*p* > 0.05). Furthermore, there were no differences (*p* > 0.05) in otological symptoms (tinnitus, sudden unilateral loss of hearing, progressive loss of hearing) in female patients. Unfortunately, the proportion of female patients was too small to facilitate statistically significant results. According to Cohen’s classification, the effect sizes observed in the statistically significant comparisons can be considered small effect sizes, which results from large disparity between group sizes.

## 4. Discussion

ENT symptoms were frequently observed in COVID-19 infection; however, much variability has been reported in the literature. El-Anwar et al. reported the most common manifestations to be cough (63.3%), dyspnea, (45%) sore throat (30%), nasal congestion (28.3%), nasal obstruction (26.7%), sneezing (26.6%), headache (25%) and smell/taste dysfunction (25%) [3]. However, another study by the same author presented sore throat (11.3%) and headache (10.7%) as the most frequently occurring symptoms [2]. Zięba et al. pointed out olfactory disorders (72%), taste disturbance (68%), and vertigo and dizziness (34%) [4]. Korkmaz et al. mentioned sore throat and smell/taste disturbances as prevailing symptoms [5]. We assume that results differed substantially between studies due to various factors influencing the tested groups. Possible factors might be age, comorbidities, research unit (blood donation center or ENT clinic), and geographical region, which could be associated with non-specific mutations of SARS-CoV-2. Furthermore, the ENT symptoms that the authors considered influenced the study results.

Our study showed no significant differences in ENT symptom manifestation in younger patients as compared with older patients, except for smell/taste disturbance, which occurred more often in patients under 39.8 years of age (Table 2). Elibol et al. report that otolaryngological symptoms are more frequent in the 18–30 age group, which is consistent with our smell/taste disturbance results **.** The authors noted that ENT manifestations of COVID-19 occurred more often in women [16]. Moreover, Takahashi et al. found gender differences in immune responses to SARS-CoV-2 and in prognostic factors for disease progression [17]. However, in our research, symptoms had different frequencies, e.g., sore throat occurred more often in women (31.82%) than men (23.84%), yet statistical significance was not achieved. As such, we are unable to conclude from our data that gender affects otolaryngological symptoms (Table 2). Furthermore, the female patient group in our study was small, probably caused by the exclusion criteria, as HLA antibodies in the blood appear after pregnancy. Therefore, statistical significance could be underestimated.

This study assessed otolaryngological symptom correlation with the level of anti-SARS-CoV-2 antibody titers; to the authors’ knowledge, this is the first study to investigate this. This work may give rise to new considerations if ENT symptom occurrence could become a predictor of immune response; however, this requires further research with larger groups of people.

Plasma substance concentration could have a significant impact on the body’s functioning; e.g., potassium disorders can lead to life-threatening conditions [18]. Zheng et al. measured serum albumin levels in patients with sudden sensorineural hearing loss (SSNHL) and proved low albumin levels to be associated with the worse SSNHL functional outcome [19]. We expect that plasma anti-SARS-CoV-2 antibody titer could contribute as a predictive factor for disease severity, duration, and complications.

Convalescent plasma is an accepted method of severe SARS-CoV-2 treatment [20,21]. Song et al. noted that the month after SARS-CoV-2 infection was the time of highest immunoglobulin titer [6]. Another study showed that the persistence of antibodies in plasma after COVID-19 infection was a minimum of 39 weeks [22]. Research has shown that administering convalescent plasma reduced mortality, the need for intubation, and the length of hospitalization [7,23]. However, complications of convalescent plasma immunotherapy, such as transfusion-related acute lung injury (TRALI), were reported [24]. After COVID-19 vaccination was introduced, it became the main method of prevention and thus also of fighting the virus. However, it did not completely eliminate the symptomatic course of the disease. Higher antibody titers post-vaccination were associated with worsening COVID-19 symptoms, which is likely due to an excessive immune response [25]. Every treatment has its side effects and possible complications, thus requiring an individualized approach and patient observation.

## 5. Conclusions

The most common ENT symptoms of SARS-CoV2 infection are smell/taste disturbances, dry cough, sore throat, and dyspnea. Smell or taste disturbances more often occur in younger patients and are related to lower anti-SARS-CoV-2 antibody titers in convalescent plasma. This study showed a statistically significant correlation between the occurrence of some otolaryngological symptoms and anti-SARS-CoV-2 antibody levels in convalescent plasma. Gender does not affect the occurrence of ENT symptoms during COVID-19 infection.

## Figures and Tables

**Figure 1 jcm-12-00610-f001:**
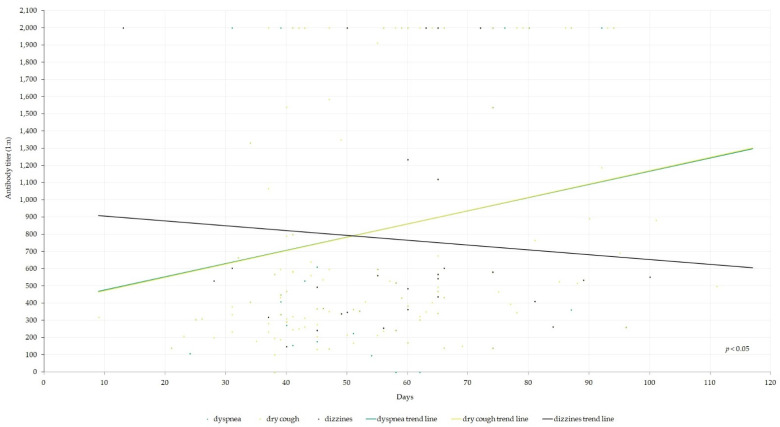
Anti-SARS-CoV-2 antibody titers depending on the symptoms of dizziness, dyspnea, and dry cough. Spots may represent more than one symptom. The chi-square test was used to derive the *p*-values.

**Figure 2 jcm-12-00610-f002:**
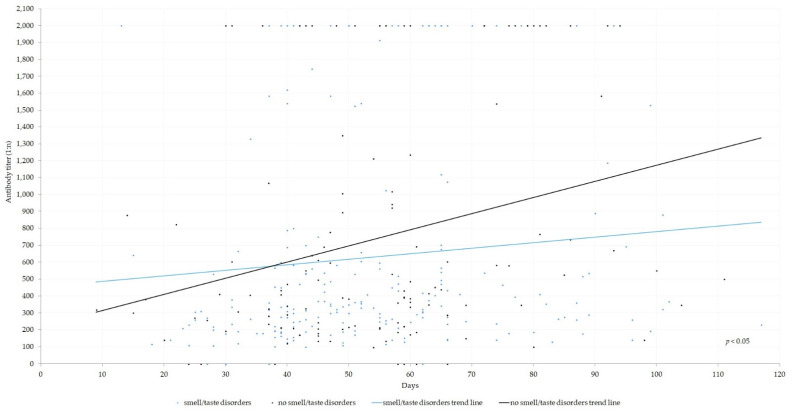
Anti-SARS-CoV-2 antibodies among patients depend on smell/taste disturbances. The chi-square test was used to derive the *p*-values.

**Figure 3 jcm-12-00610-f003:**
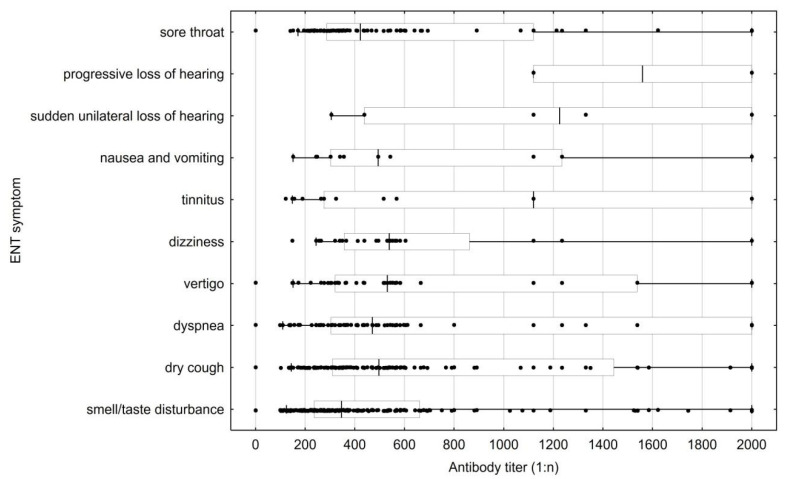
Antibody titer by reported ear, nose, and throat (ENT) symptom. Values for individuals with multiple symptoms are shown for each symptom individually. The center line denotes the median value (50th percentile), the box contains the 25th to 75th percentiles of the dataset, the whiskers mark the 5th and 95th percentiles, and values beyond these upper and lower bounds are considered outliers.

**Table 1 jcm-12-00610-t001:** Demographic data.

Index	N
Mean age (95% CI)	39.80 (38.79–40.81)
Gender	
Male	302
Female	44
Antibody titer	
<1:368.5 (male:female ratio)	6:1
>1:368.5 (male:female ratio)	9:1
Clinical information	
Hypertension	9 (only males)
Familial hypercholesterolemia	1 (only females)
No comorbidities (male:female ratio)	7:1
Ethnicity	
Polish (%)	346 (100%)

**Table 2 jcm-12-00610-t002:** Statistical analysis of ear, nose, and throat (ENT) ENT symptoms by antibody titer; sex, and (age. The chi-square test and the Fisher exact test were used to derive the *p*-values; *p* < 0.05—outcome statistically significant (marked in bold); Cohen’s h effect size was presented; 1:368.5—median antibody titer.

ENT Symptom	<1:368.5 (N = 173)	>1:368.5 (N = 173)	*p* Value	Cohen’s h	Males (N = 302)	Females (N = 44)	*p* Value	Cohen’s h	Age < 39.8 (N = 167)	Age > 39.8 (N = 179)	*p* Value	Cohen’s h
smell/taste disturbance	117	99	**0.046 ^a^**	0.215	184	32	0.131 ^a^	0.252	119	97	**0.001 ^a^**	0.355
dry cough	51	89	**<0.0001 ^a^**	0.452	125	15	0.357 ^a^	0.151	65	75	0.573 ^a^	0.061
sore throat	39	47	0.32 ^a^	0.107	72	14	0.253 ^a^	0.178	38	48	0.382 ^a^	0.094
dyspnea	30	52	**0.005 ^a^**	0.302	72	10	0.871 ^a^	0.026	45	37	0.17 ^a^	0.148
vertigo	15	26	0.067 ^a^	0.198	37	4	0.544 ^a^	0.103	19	22	0.793 ^a^	0.028
dizziness	8	20	**0.018 ^a^**	0.26	26	2	0.554 ^b^	0.166	15	13	0.558 ^a^	0.063
tinnitus	8	13	0.26 ^a^	0.122	19	2	1 ^b^	0.077	8	13	0.336 ^a^	0.104
nausea and vomiting	6	7	0.786 ^a^	0.03	10	3	0.221 ^b^	0.162	6	7	0.885 ^a^	0.017
sudden unilateral loss of hearing	1	5	0.215 ^b^	0.189	6	0	1 ^b^	0.283	2	4	0.686 ^b^	0.081
progressive loss of hearing	0	2	0.499 ^b^	0.215	2	0	1 ^b^	0.163	1	1	1 ^b^	0.005

^a^*p*-value calculated with chi-squared test, ^b^*p*-value calculated with Fisher exact test.

## Data Availability

No public database has been created. All data are available from the authors of the work.

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
