# Peer review of "Correlation of ENT Symptoms with Age, Sex, and Anti-SARS-CoV-2 Antibody Titer in Plasma"

_jcm, 2023, doi:10.3390/jcm12020610_

Round 1
Reviewer 1 Report (Previous Reviewer 2)
Thank you for incorporating the suggested changes/edits and significantly improving on the manuscript. I have no further comments/suggestions to make.
Author Response
We kindly thank you for your time and review.
Reviewer 2 Report (New Reviewer)
I think the paper overall provides valuable results regarding the correlation of ENT symptoms and the SARS-CoV-2 antibody titers. I have several questions and comments, and I believe this manuscript can be accepted after minor revisions.
- I am little unsure whether using median to divide the group is a better choice. Since the mean value (i.e., 1:400) is within the FDA standard level (i.e., 1:160–1:640), using the mean value instead seems to be more logical. The authors argued that they used median to adjust outlying values. I wonder if removing outliers at all makes more sense. Showing the detailed distribution of antibody titers using a plot (e.g., histogram) will be also helpful.
- Sample size does not seem to be sufficiently large to get meaningful statistical results when each group becomes to have too small samples (e.g., tinnitus). At least authors need to consider showing data visually (e.g., box plots as in Fig. 1A [&]) to make the results more readable and interpretable.
- Figure 1 and 2 can be improved. The font size should be increased for the readability, and use colorblind-friendly colors (e.g., search for the Okabe-Ito color scheme). Showing grid lines inside plots would be also helpful.
- The percent of vertigo is omitted when describing Zięba et al.'s work ([4] in the manuscript) in Discussion.
[&]: https://www.nature.com/articles/s41467-021-21336-8
Author Response
Please see the attachment.

This manuscript is a resubmission of an earlier submission. The following is a list of the peer review reports and author responses from that submission.
Round 1
Reviewer 1 Report
The abstract does not reflect the results obtained.
The introduction is too general, very brief. It does not describe what has been researched so far.
This type of written introduction is completely unpublishable. There are many similar articles that the authors have not considered. It is not known what is the innovation of this study (if it exists).
Statistical analysis is negligible. Relevant statistical analyzes were not performed. There are no relevant correlation analyzes and intergroup comparisons. More advanced statistical analyzes should be applied. At the moment, the results are of very low quality.
The exclusion criteria were not adequately described.
The effect size for the statistical test used was not investigated.
The results of the statistical tests are not recorded according to scientific standards.
Reviewer 2 Report
This paper reports the correlation between ENT symptoms occurrence and antibody titer in convalescent plasma from a cohort of 346 COVID-19 convalescents who donated blood in Gdansk, Poland. While this is an interesting study, there are several major points and issues with the analyses that need to be addressed.
1. Please summarize the inclusion and exclusion criteria, blood and serological tests in this paper, rather than just referencing a previous paper and expecting the reader to get this important information from a separate publication.
2. Could the authors please provide more details on the demographic of the patients, in particular ethnicity and presence or absence of comorbidities? Perhaps include a table summarizing the patient demographics.
3. What was the age range of the patients? What do the authors mean by ‘average age’? Is this mean or median? Why did the authors choose ‘average age’ as the cut off to divide patients into younger and older age groups? An age of 38.9 is not what one might usually consider as ‘old’.
4. Why was the median antibody titre of 1/368.5 used as a cut-off between high and low antibody titres? This seems rather low compared to the cut-off levels for high titre (1/500 – 1/600) used in other studies (e.g. https://doi.org/10.1093/cid/ciaa721 and https://doi.org/10.1101/2022.04.29.22274387).
5. Statistical analysis: Was post hoc Bonferroni correction applied to their Chi-squared test to correct for multiple comparisons? Otherwise, there might be false positives from their statistical analysis.
6. Figure 1 and Figure 2: how were the trend lines generated? There seems to be a large overlap in the scatter of the individual datapoints. What statistical test were used to generate the p value in these figures?
Minor comment: while most part of the paper reads well, some parts need English language editing with improper use of certain words e.g. in the abstract ‘We marked the level of antibodies…’ should be ‘We measured the level of antibodies…”.
Round 2
Reviewer 1 Report
Statistical analysis is still at a very low level.
The statistical tests used were not described.
The size of the effect as measured by the respective coefficients has not been calculated.
It is not known where and which statistical test was used in the results.
“We provided a post-hoc Bonferroni test to prevent data from incorrectly appearing to be statistically significant” - This statistical test is used for other purposes.
The results of statistical tests are still not recorded according to scientific standards. For example: Chi Square Chi-Square statistics are reported with degrees of freedom and sample size in parentheses, the Pearson chi-square value (rounded to two decimal places), and the significance level.
The abstract does not reflect the results obtained (significance level).
Reviewer 2 Report
Thank you for the opportunity to read the revised version of this work. Overall, the authors made significant changes to their manuscript and I am grateful for that. However, there are still some minor issues in their revised version:
Spelling errors: there are still several spelling errors throughout the manuscript (e.g. 'mesured', 'assosiated', hypercholesterolemia (in table 1), etc.). May I suggest the authors use a spell-checker?
Table 1: Ethnicity should be 'Polish' or 'Pole', rather than 'Polen'.
Response to Point 4 - I am satisfied with the explanation given. However, please can the authors mention their justification in the manuscript?
Response to Point 5 - thank you for calculating the values with post hoc Bonferroni correction. Please can the authors state in their manuscript that post hoc Bonferroni correction was applied to their analyses?
Response to Point 6 - please can the authors state in their manuscript the statistical test (Chi square and Bonferroni correction) used to derive the P values.
In the Discussion, the authors write "However, in our research, symptoms had different frequencies, e.g., sore throat occurred more often occurred in women (31.82%) than men (23.84%), yet statistical significance was not achieved. As such, it was assumed that assumed gender does not affect otolaryngological symptoms". It would not be accurate to say that "gender does not affect otolaryngological symptoms" is an assumption simply because statistical significance is not achieved.
May I suggest the authors reword it to say "As such, we are unable to conclude from our data that gender affects otolaryngological symptoms."
